# Analysis of the Active Measurement Systems of the Thoracic Range of Movements of the Spine: A Systematic Review and a Meta-Analysis

**DOI:** 10.3390/s22083042

**Published:** 2022-04-15

**Authors:** Pablo Esteban-González, Eleuterio A. Sánchez-Romero, Jorge Hugo Villafañe

**Affiliations:** 1Department of Physiotherapy, Faculty of Sport Sciences, Universidad Europea de Madrid, 28670 Vil-laviciosa de Odón, Madrid, Spain; 2Musculoskeletal Pain and Motor Control Research Group, Faculty of Sport Sciences, Universidad Europea de Madrid, 28670 Villaviciosa de Odón, Madrid, Spain; 3Department of Physiotherapy, Faculty of Health Sciences, Universidad Europea de Canarias, 38300 La Orotava, Canary Islands, Spain; 4Musculoskeletal Pain and Motor Control Research Group, Faculty of Health Sciences, Universidad Eu-ropea de Canarias, 38300 La Orotava, Canary Islands, Spain; 5IRCCS Fondazione Don Carlo Gnocchi, Piazzale Morandi 6, 20141 Milan, Italy

**Keywords:** range of motion, movement, mobility, range of movement, thoracic, spine, system, device, tool

## Abstract

(1) Objective: to analyze current active noninvasive measurement systems of the thoracic range of movements of the spine. (2) Methods: A systematic review and meta-analysis were performed that included observational or clinical trial studies published in English or Spanish, whose subjects were healthy human males or females ≥18 years of age with reported measurements of thoracic range of motion measured with an active system in either flexion, extension, lateral bending, or axial rotation. All studies that passed the screening had a low risk of bias and good methodological results, according to the PEDro and MINORS scales. The mean values and 95% confidence interval of the reported measures were calculated for different types of device groups. To calculate the differences between the type of device measures, studies were pooled for different types of device groups using Review Manager software. (3) Results: 48 studies were included in the review; all had scores higher than 7.5 over 10 on the PEDro and MINORs methodological rating scales, collecting a total of 2365 healthy subjects, 1053 males and 1312 females; they were 39.24 ± 20.64 years old and had 24.44 ± 3.81 kg/m^2^ body mass indexes on average. We summarized and analyzed a total of 11,892 measurements: 1298 of flexoextension, 1394 of flexion, 1021 of extension, 491 of side-to-side lateral flexion, 637 of right lateral flexion, 607 of left lateral flexion, 2170 of side-to-side rotation, 2152 of right rotation and 2122 of left rotation. (4) Conclusions: All collected and analyzed measurements of physiological movements of the dorsal spine had very disparate results from each other, the cause of the reason for such analysis is that the measurement protocols of the different types of measurement tools used in these measurements are different and cause measurement biases. To solve this, it is proposed to establish a standardized measurement protocol for all tools.

## 1. Introduction

Over the years, different types of spinal column tools have been appearing; some are directed to the cervical spine, others to the lumbar spine, and many others to the dorsal spine. These tools are essential for the assessment of the column, serve as a method of evaluating joint mobility [1], assess the existence of pain on movement [2,3], and prevent vertebral pathologies [4,5,6]. They are also essential for the research that has been carried out through the years on new spinal treatment methods.

The physiological movements of the dorsal spine have been measured for more than 40 years, leading to the emergence of a multitude of tools and different performance protocols from the first studies with goniometers and inclinometers [7] to the latest with applications for cell phones [8]; within this large section, the tools can measure one or more of the different planes of motion that the dorsal spine has, which are the sagittal plane (flexion and extension), the coronal plane (right tilt and left tilt) and the transverse plane (right rotation and left rotation). 

Within the wide variety of tools, a distinction can be made between different types depending on how they obtain the degrees of measurement. The first tools to emerge were mechanical devices [8,9,10]. They are those tools not provided with electricity that work by transmitting the movement of the subject, measuring the degrees of mobility directly. These devices include, for example, the Goniometer and the Inclinometer. The rest of the tools were emerging with the industrial evolution and the improvement of the present resources, as was the case of the electromechanical devices [11,12,13] that are simply mechanical tools equipped with better and more innovative electronic equipment to improve the sensitivity, specificity, and comfort of the professional by increasing the complexity of the tool and providing it with a receiver responsible for transmitting the movement of the subject to be processed by a computer. These tools are the Electro-Goniometer, the Digital Inclinometer, and the Spinal Mouse. On the other hand, another way of assessing dorsal physiological movements through the study of images emerged; these tools were called three-dimensional optical motion analysis devices [13,14]. They consist of a photo or video camera and an image analysis processor studying either the image of the initial position and the image of the final position or the elapsed movement, in other words, these tools analyze three-dimensional images. As new methods of reception and transmission of dorsal physiological movements emerged, new tools appeared, such as accelerometer tracking device [15] (XSENS TMX or the 3A Sensor String), ultrasound tracking device [16] (CMS 20 ZEBRIS), and electromagnetic tracking device [17] (Fastrak), these tools obtain the degrees of movement of the subject with a specialized sensor. Finally, the last type of tools were adapted to our daily life, with phones used for the measurement of dorsal ranges; these tools are mobile phone applications [8,11], as is the case of the Clinometer App. The classification of all the tools analyzed in this study can be seen in Table 1.

In addition to the different types of tools present, it is also important to know the different measurement positions that are present within the protocols of each tool, which are seated, standing, gentleman position, and Mahometan position.

Throughout the years that these tools have been emerging, there has been no consensus, in fact, each tool has its initial position, its final position, its placement of the device, its different indications, its measurement time, in short, each tool has its protocol of action. It is interesting to see how these factors can affect the measurements of the different movements of the dorsal column; therefore, we aimed to analyze current active noninvasive measurement systems of the thoracic range of movements of the spine.

## 2. Methods

The systematic review realized in this study is in accordance with the PRISMA statement extension for systematic reviews incorporating network meta-analysis: PRISMA-NMA statement (2015) [18] and with The PRISMA 2020 statement: an updated guideline for reporting systematic reviews [19]. This study followed PROSPERO [20] regulations and guidelines and is registered under ID CRD42021231380.

### 2.1. Eligibility Criteria

The following inclusion criteria were used: (1)measurements were performed on asymptomatic subjects without a current or previous history of spinal disorders or low back pain, (2) subjects were male or female humans ≥18 years old, (3) reported thoracic RoM measurements, (4) measured an active RoM either in flexion, extension, lateral bending, or axial rotation, (5) referenced initial position measurement, (6) published observational or clinical trial studies and (7) studies in English or Spanish.

On the other side, the studies with the following standards were excluded: (1) if the studies were Review, Meta-analysis, Case reports, Systematic review, book or letter, (2) any cadaveric or impact studies, (3) if subjects had any pathology or surgery of the spine, cancer, aorta’s pathology, rheumatic diseases, or scoliosis, (4) if the measurements were made with ionizing devices or with tools that there were no records of their use in the last 20 years, (5) studies of the respiratory movements and (6) studies of the thoracic spine movement while walking or running.

### 2.2. Information Sources and Search Strategy

The search began on Monday 11 January 2021 and finished on Friday 17 February 2021. Search criteria with MESH terms, including Thoracic, tool, device, system, measure, rotation, bending, extension, flexion, motion, mobility, kinematic, movement and range of motion, were used with logical operators (AND, OR) to search the electronic databases of (1) PubMed (National Library of Medicine and National Institute of Health, Bethesda, MD, USA), (2) Cochrane (Clarivate analytics, USA), (3) EMBASE (Elsevier, Amsterdam, The Netherlands, NLD) and (4) Web of Science (Clarivate analytics, USA). The full search string that was used is in Appendix A.

### 2.3. Selection Process

All the selection process was made by one of the authors (P.E.-G.). In the initial search, it was utilized an automatic tool of the databases to remove all the Review, Meta-analysis, Case reports, Systematic review, book, or letter study types. After that, studies passed the initial screening by titles, screening by abstract, and concluded with a full-text screening following the inclusion and exclusion criteria mentioned above.

### 2.4. Data Collection Process and Variables

The data were managed in the Microsoft Excel^®^ software, where the extraction data including the year of publication, number of subjects, sex, body height, weight, body mass index, name device, type of measuring device, software device, measurement posture, and the right and left thoracic RoM of flexion, extension, lateral bending or axial rotation measured, were tabulated. We recruited all of these measures of each device.

### 2.5. Study Risk of Bias Assessment

Initially, two authors (P.E.-G. and E.A.S.-R.) collected the papers included in the review and studied the methodological quality and risk of bias of the articles using the PEDro scale for experimental studies and MINORS scale for observational studies.

The PEDro scale is based on the Delphi list developed by Verhagen and collaborators at the Department of Epidemiology, Maastricht University [21]. For the most part, the list is based on expert consensus and not on empirical data. The purpose of the PEDro scale is to help users of the PEDro databases to quickly identify which of the randomized clinical trials may have sufficient internal validity (criteria 2–9) and sufficient statistical information to make their results interpretable (criteria 10–11). An additional criterion (criterion 1) relates to external validity, but this criterion will not be used for the calculation of the PEDro scale score.

The MINORS Scale [22,23] (Methodological index for non-randomized studies) is a tool designed to evaluate non-randomized trials and observational studies. This scale includes 8 items for non-randomized studies and 4 more items for comparative studies. Each item is evaluated with a score between 0 (not reported), 1 (reported but incomplete) and 2 (reported and complete). The first 4 items plus the 8th item refer to the methodology and design of the study, whereas items 5, 6, 7 refer to the results obtained. On the other hand, items 9, 10, 11 and 12 are based on additional criteria for comparative studies. The maximum score that can be obtained is 16 for non-randomized studies and 24 for comparative studies.

### 2.6. Data Synthesis Methods and Meta-Analysis

The mean values and 95% confidence interval (CI) range of Flexo-Extension (rFE), range of Flexion (rF), range of Extension (rE), range of Side to Side Lateral Flexion (rSSLF), range of Right Lateral Flexion (rRLF), range of Left Lateral Flexion (rLLF), range of Side to Side Rotation (rSSR), range of Right Rotation (rRR) and range of Left Rotation (rLR) were calculated for different type of device groups: Mechanical Devices (MD), Electro Mechanical Devices (EMD), 3Dimensional Optical Motion Analysis (3-DOMA), Accel-erometer Tracking Devices (ATD), Ultrasound Tracking Devices (UTD), Electro Magnetic Tracking De-vices (EMGTD) and Mobile Phone Applications (MPA). To calculate the differences between the type of device measures, studies were pooled for different types of device groups using Review Manager software (RevMan, version 5.4. Copenhagen: The Nordic Center, The Cochrane Collaboration, London, UK, 2014). The meta-analysis was performed using the random-effects model to compare the different types of devices measures and for considering heterogeneity among all measures. Statistical heterogeneity was evaluated based on the inconsistency (I2) index that provides an estimated percentage of the total variation across the measures of the studies that were included. The scale of heterogeneity was considered, whereby <25% indicates low, 25–75% medium, and >75% high heterogeneity [24]. Mean pooled differences and 95% CIs in rFE, rF, rE, rSSLF, rRLF, rLLF, rSSR, rRR, and rLR between different types of device measures were presented as statistically significant when *p* < 0.05.

## 3. Results

### 3.1. Study Selection

The electronic search saved a total of 27,266 publications in the databases mentioned (Figure 1). Following the selected criteria, title and abstract screening, and the removal of the duplicates in the four databases, 27,106 papers were excluded. The remaining 160 full-text papers were screened for eligibility and 112 were excluded. Among these, 78 had missing thoracic measurements, 9 had subjects with pathology, 8 had subjects aged under 18 years, 7 studies were in Chinese, German, or Polish, 5 papers had cadaveric or scoliosis measurements, 3 publications had measurements made with ionizing devices and 2 studies had measurements made with tools that there were not records of their use in the last 20 years. Finally, 48 studies were included in the review; all had scores higher than 7.5 over 10 on the PEDro and MINORs methodological rating scales (Appendix B and Appendix C). These studies were performed in Europe (17), Oceania (11), Asia (11), and Americas (9). Among these, 48 studies were considered in the meta-analysis.

### 3.2. Study Characteristics

A synthesis of the objective, methodology, and results of the included studies are presented in Table 2 and their characteristics are presented in Table 3. The total number of healthy participants in these papers was 2365, with 1053 males and 1312 females; the sample size ranged from 12 [25,26] to 120 [27], in the sagittal plane. The total number of healthy measured participants was 1707, the total in the coronal plane was 591, and in the transversal plane, the total number was 888 subjects. The number of healthy female and male subjects are presented in Table 4.

A total of seven types of the device were registered: mechanical devices (42 measures), electromechanical devices (34 measures), three-dimensional optical motion analysis (46 measures), accelerometer tracking devices (20 measures), ultrasound tracking devices (20 measures), electromagnetic tracking devices (12 measures) and mobile phone applications (9 measures). All measurements realized according to the different ranges of movement and the device used are present in Table 5. The EMD and the UTD were not used to measure the coronal plane and the MPA were not used to measure the sagittal and coronal plane.

Three different types of postures were collected in the measurements: standing (32 measures), sitting (18 measures), and lumbar locked rotation test (10 measures). The devices that made the most measurements while standing (12) were electromechanical devices and three-dimensional optical motion analysis. The device that made the most measurement while sitting and in the lumbar locked rotation test (5) was mechanical devices. All the measurements postures and the devices used are present in Table 6.

The demographic data of the selected subjects depending on the type of device used are presented in Table 7 and the demographic data of the selected subjects depending on the type of position measure are presented in Table 8.

Grouping tools according to the type of device we summarized and analyzed a total of 11,892 measurements. Of these, 3713 were from the sagittal plane: 1298 of flexoextension, 1394 of flexion, 1021 of extension. The differences in flexoextension were those shown in Figure 2, Figure 3, Figure 4, Figure 5, Figure 6 and Figure 7, the differences in flexion were those shown in Figure 8, Figure 9, Figure 10, Figure 11, Figure 12 and Figure 13 and the differences in extension were those shown in Figure 14, Figure 15, Figure 16, Figure 17, Figure 18 and Figure 19.

Of 11,892 measurements, 1735 were from the coronal plane: 491 of side-to-side lateral flexion, 637 of right lateral flexion, 607 of left lateral flexion. The differences in side-to-side lateral flexion were those shown in Figure 20, Figure 21, Figure 22 and Figure 23, the differences in right lateral flexion were those shown in Figure 24, Figure 25, Figure 26 and Figure 27, and the differences in left lateral flexion were those shown in Figure 28, Figure 29, Figure 30 and Figure 31.

A total of 6444 measurements were from the transversal plane: 2170 of side to side rotation, 2152 of right rotation and 2122 of left rotation, the differences in side to side rotation were those shown in Figure 32, Figure 33, Figure 34, Figure 35, Figure 36, Figure 37 and Figure 38, the differences in right rotation were those shown in Figure 39, Figure 40, Figure 41, Figure 42, Figure 43, Figure 44 and Figure 45 and the differences in left rotation were those shown in Figure 46, Figure 47, Figure 48, Figure 49, Figure 50, Figure 51 and Figure 52.

Grouping tools according to the type of position measure we summarized and analyzed a total of 12,092 measurements. Of these, 3713 were from the sagittal plane and 1935 from the coronal plane. The differences in flexoextension, flexion, extension, side to side lateral flexion, right lateral flexion, and left lateral flexion, were those shown in Table 9.

In the transversal plane, we analyze a total of 6444 measurements, the differences in side-to-side rotation were those shown in Figure 53, Figure 54, Figure 55 and Figure 56, the differences in right rotation were those shown in Figure 57, Figure 58, Figure 59 and Figure 60 and the differences in left rotation were those shown in Figure 61, Figure 62, Figure 63 and Figure 64.

## 4. Discussion

The current study aimed to analyze the different tools used nowadays to measure the physiological movements of the dorsal spine reviewing their different protocols and measurements.

Interpreting the results of the meta-analyses of the measurements based on the type of measurement tool used, the values obtained from the flexo-extensions, the measurement grades of the MDs and ATDs presented significant differences for the rest of the measuring devices used, whereas the measurement grades of the EMDs and 3-DOMA only had significant similar results concerning each other. Similarly, UTD and EMGTD values were only significantly similar to each other. When observing the results obtained in flexion, the MD grades had similar significant results with the values of the 3-DOMA and ATD, the rest of the values were significantly different. The EMD values only had significantly similar results with the ATD, similarly, the UTD and EMGTD values only had significantly similar results with each other. Finalizing the values of the sagittal plane, in the extension values, the MDs measurements had similar significant results with the ATD, UTD, and EMGTD. The results measured with the EMD were only significantly similar to the values of the ATD, whereas the values of the 3-DOMA had significant differences with the rest of the measuring devices used. The ATDs had similar results with the MD, EMD, UTD, and EMGTD, the values obtained with the UTD in addition to having similar significant results with the MD and ATD also had them with the EMGTD measurements. In all the values of the coronal plane movements (side-to-side lateral flexion, right lateral flexion, and left lateral flexion) obtained with the different types of tools, there were significant differences, except with the values obtained in the left lateral flexion with the MD and ATD, which had similar significant results. In the transverse plane, in the values obtained in the side-to-side rotation, the measurements obtained with the MDs had similar significant results with the ATDs and MPAs, on the other hand, the values of the EMDs were significantly similar only to the STDs. The 3-DOMA values were not significantly similar to the rest of the tool types. The ATD had significantly similar results with the MD and MPA, the EMGTD only had significantly similar measurements with the UTD. On the other hand, in the values obtained in the right rotation measurements, the MD had similar significant results with the EMD, ATD, and MPA, and it was reciprocal in the values of the EMD, ATD, and MPA. Similar to the side-to-side rotation measurements, the 3-DOMA values were not significantly similar to the other tool types. The UTD and EMGTD measurements only had significantly similar results with each other. The similarities in the left rotation measurements obtained with the different tool types were similar to those of the right rotation, except that the 3-DOMA values were significantly related to those of the ATDs.

Interpreting the results of the meta-analyses of the measurements based on the type of posture, all movements measured were significantly different, except for right and left rotations measured in the seated and locked lumbar postures, which had similar significant results, this indicates that the positions adopted in the measurements are a key factor that makes the measurements differ from each other even when measuring the same movements.

Another reason that shows that the results were very different from each other is the I2, since all the results were around 75%, which shows that in the comparisons of the different physiological movements of the dorsal spine according to the position and type of tool, the measurements were very irregular from each other, having a large statistical heterogeneity.

Most of the measurements of the different physiological movements of the dorsal spine depending on the type of instrument used, which should give similar results, vary in a high percentage since it has been shown that in a statistically significant way there are considerable differences between many of them, being the measurements very irregular; even higher is the difference depending on the type of position used at the time of measurement. We have been able to verify that there are a large number of tools to measure the physiological movements of the spine, each of these tools has its different protocol and establishes its initial positions that the subjects must have to perform the measurement. Even so, if the tools were as sensitive and specific as possible, there would be no differences between them, what we have been able to verify through this study is that there are, so we should act based on this and think together about possible solutions to this problem.

No tool was found that measures the different planes of movement of the cervical, dorsal and lumbar regions. Of all the tools referenced, only some 3-DOMA [25,26,64,67], two ATD [53,62] and one EMGTD [30,35,48] (the FASTRAK) measured the three planes in the dorsal spine, so the presence of a tool capable of measuring flexion-extensions, lateral inclinations and rotations are not very common either.

The main long-term objective to solve these differences in measurements would be to perform a general protocol to try to eliminate these measurement biases but to do this, it would be important to perform a large-scale experimental study in which the physiological movements of the spine will be measured using the different types of existing tools in the same subjects. In that investigation, we will measure using the performance protocol of each tool, to corroborate the measurement differences that arise in this study and if so, to be able to establish a general protocol in which these measurement differences will be reduced.

If this is achieved, it could even be decisive in the joint assessment of the movement of the thoracic spine with the presence of pain, seeing how this variable can affect mobility, breathing and the different factors involved in the dorsal spine complex in the same way that other studies have assessed whether the presence of pain in the shoulder causes limitations in daily life [69] or whether cervical or lumbar pain can cause restrictions at work [71]. Another very interesting factor to take into account is that if we manage to make the tools as specific and sensitive as possible, they could even serve to prevent some spinal pathologies, such as osteoarthritis, by studying this pathology and checking whether they cause mobility restrictions in the initial phases.

One of the main limitations of this study is that by grouping and extracting data from the different types of tools used in the measurements of physiological movements of the dorsal spine, many studies do not distinguish between men or women, nowadays there is evidence that mobility and physiological ranges are different between male and female [71,72,73], so measurements should be discerned according to the gender of the subject.

## 5. Conclusions

The data obtained, collected and analyzed from the different physiological movements of the dorsal spine indicate that they are very irregular, depending on the type of tool used, since each of them has its action protocol. One of the most important parts of the performance protocols is the initial measurement positions adopted by the subjects. In this study, it has been shown that although the tools measure the same movement, the position adopted by the subjects ensure that the measurements do not coincide and are different. For this reason, it is important to establish a standardized performance protocol unifying initial measurement positions to try to avoid the risks of measurement bias.

## Figures and Tables

**Figure 1 sensors-22-03042-f001:**
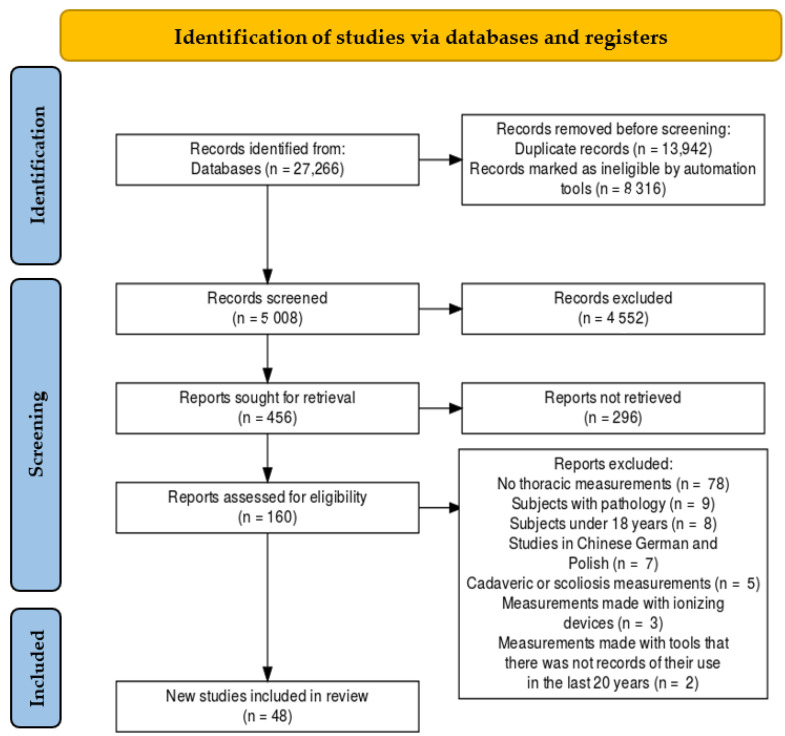
PRISMA Flow diagram.

**Figure 2 sensors-22-03042-f002:**
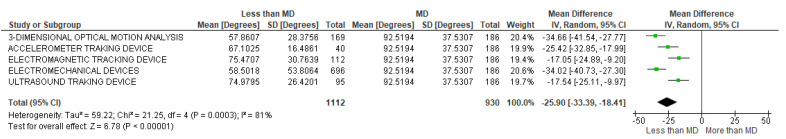
Mean, standard deviation, total of measures and 95% confidence interval comparing measured rFE of all other tool types with mechanical devices.

**Figure 3 sensors-22-03042-f003:**
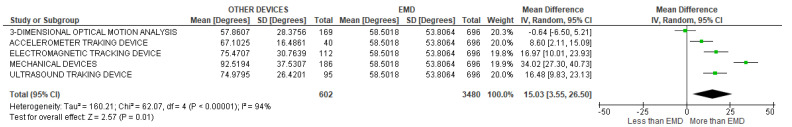
Mean, standard deviation, total of measures and 95% confidence interval comparing measured rFE of all other tool types with electromechanical devices.

**Figure 4 sensors-22-03042-f004:**
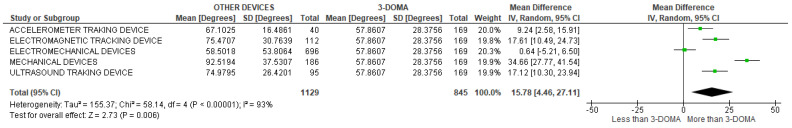
Mean, standard deviation, total of measures and 95% confidence interval comparing measured rFE of all other tool types with three-dimensional optical motion analysis.

**Figure 5 sensors-22-03042-f005:**
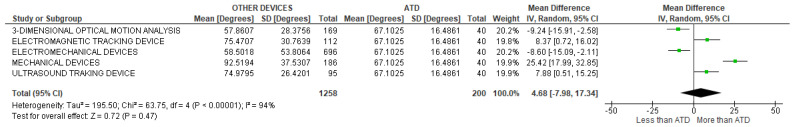
Mean, standard deviation, total of measures and 95% confidence interval comparing measured rFE of all other tool types with accelerometer tracking devices.

**Figure 6 sensors-22-03042-f006:**
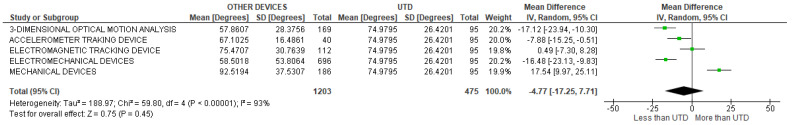
Mean, standard deviation, total of measures and 95% confidence interval comparing measured rFE of all other tool types with ultrasound tracking devices.

**Figure 7 sensors-22-03042-f007:**
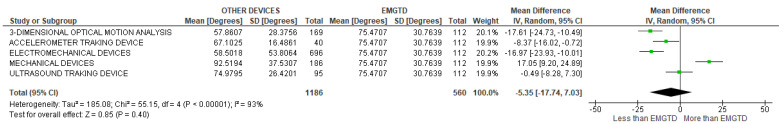
Mean, standard deviation, total of measures and 95% confidence interval comparing measured rFE of all other tool types with electromagnetic tracking devices.

**Figure 8 sensors-22-03042-f008:**
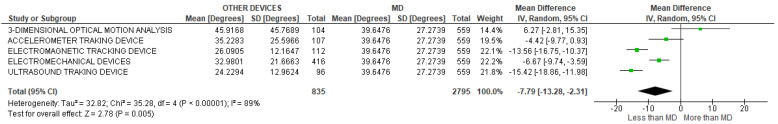
Mean, standard deviation, total of measures and 95% confidence interval comparing measured rF of all other tool types with mechanical devices.

**Figure 9 sensors-22-03042-f009:**
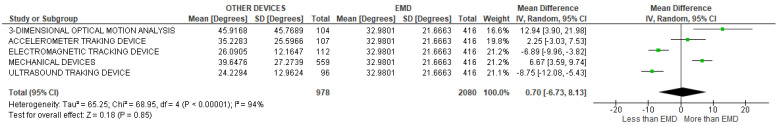
Mean, standard deviation, total of measures and 95% confidence interval comparing measured rF of all other tool types with electromechanical devices.

**Figure 10 sensors-22-03042-f010:**
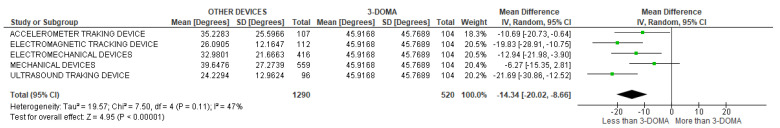
Mean, standard deviation, total of measures and 95% confidence interval comparing measured rF of all other tool types with three-dimensional optical motion analysis.

**Figure 11 sensors-22-03042-f011:**
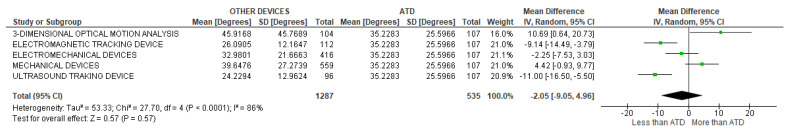
Mean, standard deviation, total of measures and 95% confidence interval comparing measured rF of all other tool types with accelerometer tracking devices.

**Figure 12 sensors-22-03042-f012:**
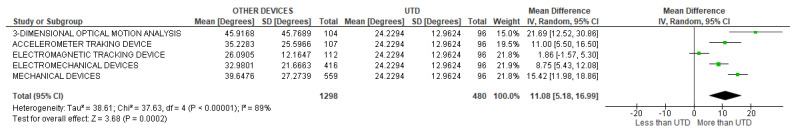
Mean, standard deviation, total of measures and 95% confidence interval comparing measured rF of all other tool types with ultrasound tracking devices.

**Figure 13 sensors-22-03042-f013:**
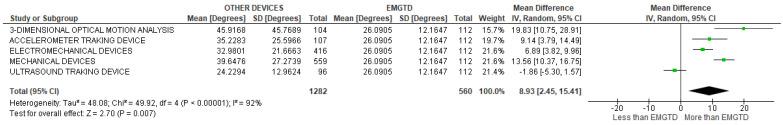
Mean, standard deviation, total of measures and 95% confidence interval comparing measured rF of all other tool types with electromagnetic tracking devices.

**Figure 14 sensors-22-03042-f014:**
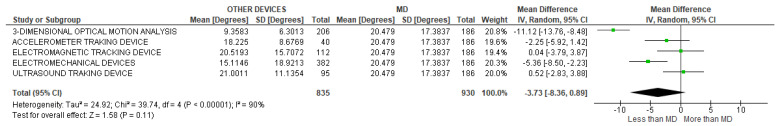
Mean, standard deviation, total of measures and 95% confidence interval comparing measured rE of all other tool types with mechanical devices.

**Figure 15 sensors-22-03042-f015:**
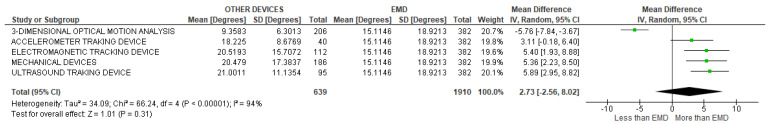
Mean, standard deviation, total of measures and 95% confidence interval comparing measured rE of all other tool types with electromechanical devices.

**Figure 16 sensors-22-03042-f016:**
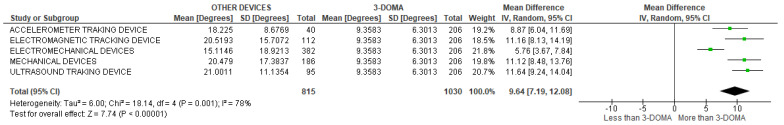
Mean, standard deviation, total of measures and 95% confidence interval comparing measured rE of all other tool types with three-dimensional optical motion analysis.

**Figure 17 sensors-22-03042-f017:**
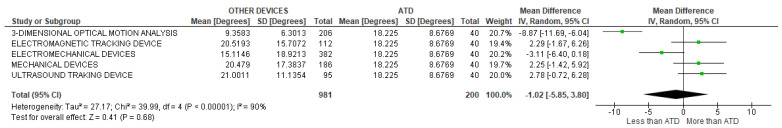
Mean, standard deviation, total of measures and 95% confidence interval comparing measured rE of all other tool types with accelerometer tracking devices.

**Figure 18 sensors-22-03042-f018:**
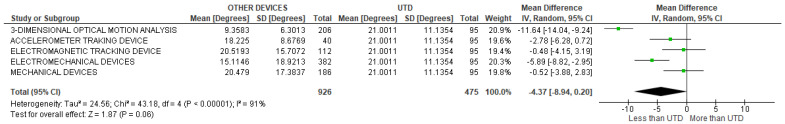
Mean, standard deviation, total of measures and 95% confidence interval comparing measured rE of all other tool types with ultrasound tracking devices.

**Figure 19 sensors-22-03042-f019:**
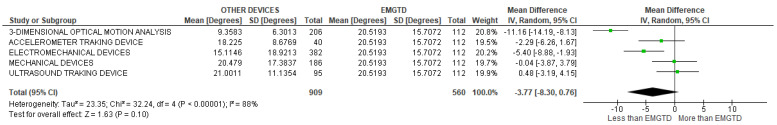
Mean, standard deviation, total of measures and 95% confidence interval comparing measured rE of all other tool types with electromagnetic tracking devices.

**Figure 20 sensors-22-03042-f020:**
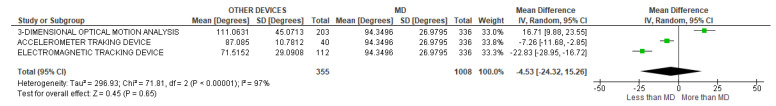
Mean, standard deviation, total of measures and 95% confidence interval comparing rSSLF of all other tool types with mechanical devices.

**Figure 21 sensors-22-03042-f021:**
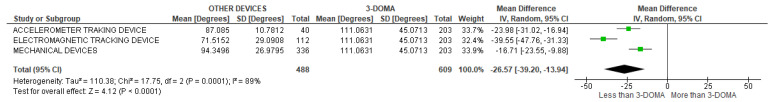
Mean, standard deviation, total of measures and 95% confidence interval comparing rSSLF of all other tool types with three-dimensional optical motion analysis.

**Figure 22 sensors-22-03042-f022:**
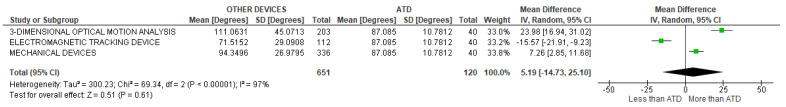
Mean, standard deviation, total of measures and 95% confidence interval comparing rSSLF of all other tool types with accelerometer tracking devices.

**Figure 23 sensors-22-03042-f023:**
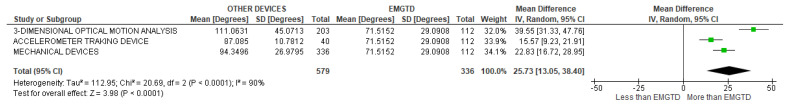
Mean, standard deviation, total of measures and 95% confidence interval comparing rSSLF of all other tool types with electromagnetic tracking devices.

**Figure 24 sensors-22-03042-f024:**
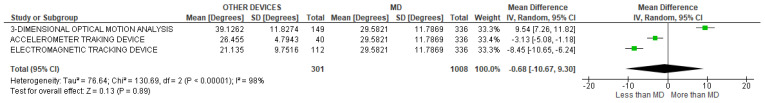
Mean, standard deviation, total of measures and 95% confidence interval comparing rRLF of all other tool types with mechanical devices.

**Figure 25 sensors-22-03042-f025:**
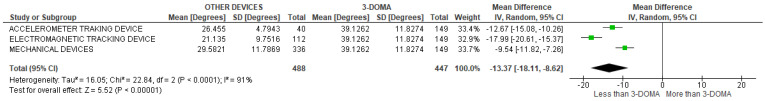
Mean, standard deviation, total of measures and 95% confidence interval comparing rRLF of all other tool types with three-dimensional optical motion analysis.

**Figure 26 sensors-22-03042-f026:**
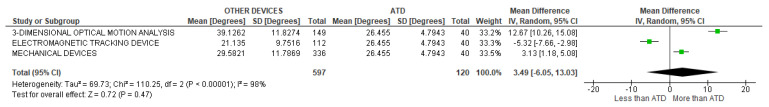
Mean, standard deviation, total of measures and 95% confidence interval comparing rRLF of all other tool types with accelerometer tracking devices.

**Figure 27 sensors-22-03042-f027:**
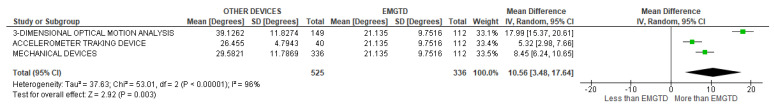
Mean, standard deviation, total of measures and 95% confidence interval comparing rRLF of all other tool types with electromagnetic tracking devices.

**Figure 28 sensors-22-03042-f028:**
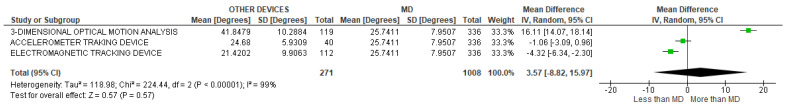
Mean, standard deviation, total of measures and 95% confidence interval comparing rLLF of all other tool types with mechanical devices.

**Figure 29 sensors-22-03042-f029:**
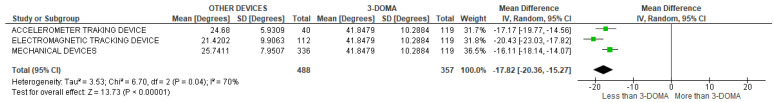
Mean, standard deviation, total of measures and 95% confidence interval comparing rLLF of all other tool types with three-dimensional optical motion analysis.

**Figure 30 sensors-22-03042-f030:**
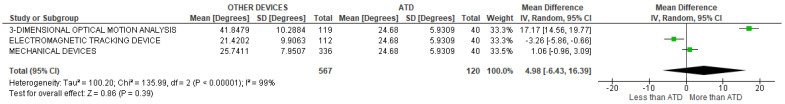
Mean, standard deviation, total of measures and 95% confidence interval comparing rLLF of all other tool types with accelerometer tracking devices.

**Figure 31 sensors-22-03042-f031:**
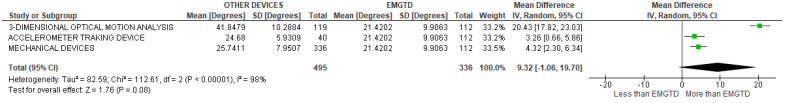
Mean, standard deviation, total of measures and 95% confidence interval comparing rLLF of all other tool types with electromagnetic tracking devices.

**Figure 32 sensors-22-03042-f032:**
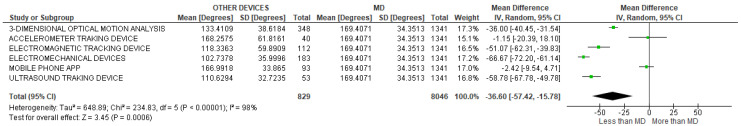
Mean, standard deviation, total of measures and 95% confidence interval comparing rSSR of all other tool types with mechanical devices.

**Figure 33 sensors-22-03042-f033:**
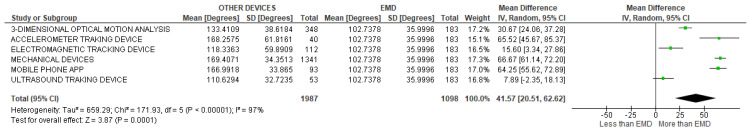
Mean, standard deviation, total of measures and 95% confidence interval comparing rSSR of all other tool types with electromechanical devices.

**Figure 34 sensors-22-03042-f034:**
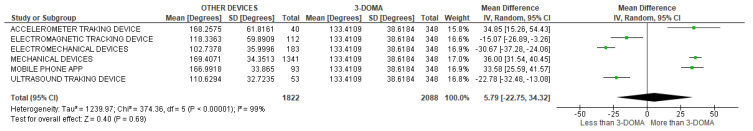
Mean, standard deviation, total of measures and 95% confidence interval comparing rSSR of all other tool types with three-dimensional optical motion analysis devices.

**Figure 35 sensors-22-03042-f035:**
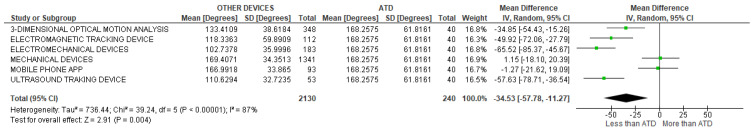
Mean, standard deviation, total of measures and 95% confidence interval comparing rSSR of all other tool types with accelerometer tracking devices.

**Figure 36 sensors-22-03042-f036:**
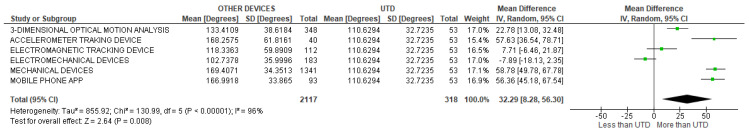
Mean, standard deviation, total of measures and 95% confidence interval comparing rSSR of all other tool types with ultrasound tracking devices.

**Figure 37 sensors-22-03042-f037:**
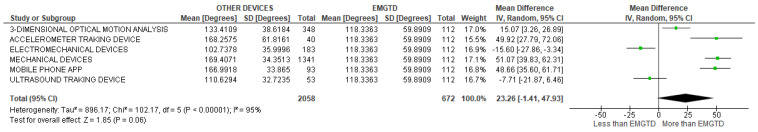
Mean, standard deviation, total of measures and 95% confidence interval comparing rSSR of all other tool types with electromagnetic tracking devices.

**Figure 38 sensors-22-03042-f038:**
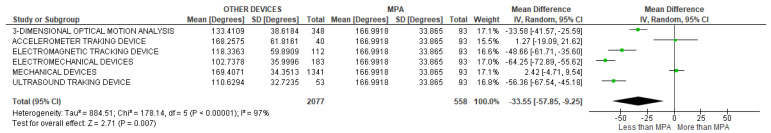
Mean, standard deviation, total of measures and 95% confidence interval comparing rSSR of all other tool types with mobile phone applications.

**Figure 39 sensors-22-03042-f039:**
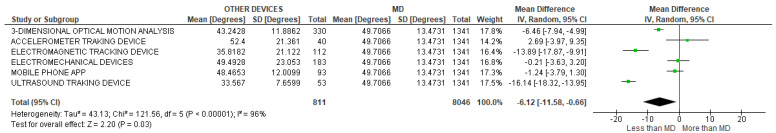
Mean, standard deviation, total of measures and 95% confidence interval comparing rRR of all other tool types with mechanical devices.

**Figure 40 sensors-22-03042-f040:**
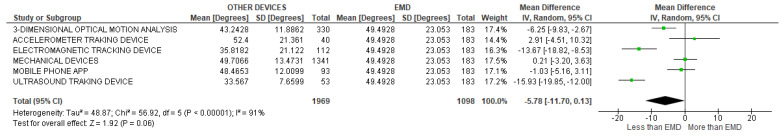
Mean, standard deviation, total of measures and 95% confidence interval comparing rRR of all other tool types with electromechanical devices.

**Figure 41 sensors-22-03042-f041:**
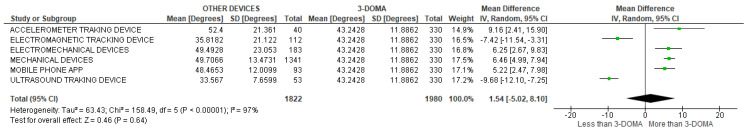
Mean, standard deviation, total of measures and 95% confidence interval comparing rRR of all other tool types with three-dimensional optical motion analysis.

**Figure 42 sensors-22-03042-f042:**
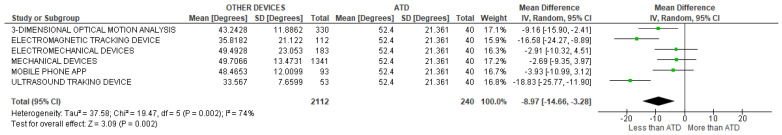
Mean, standard deviation, total of measures and 95% confidence interval comparing rRR of all other tool types with accelerometer tracking devices.

**Figure 43 sensors-22-03042-f043:**
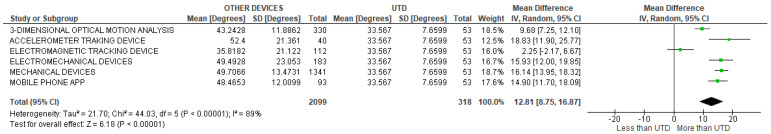
Mean, standard deviation, total of measures and 95% confidence interval comparing rRR of all other tool types with ultrasound tracking devices.

**Figure 44 sensors-22-03042-f044:**
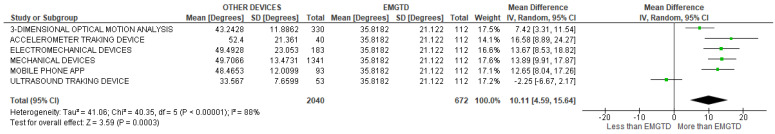
Mean, standard deviation, total of measures and 95% confidence interval comparing rRR of all other tool types with electromagnetic tracking devices.

**Figure 45 sensors-22-03042-f045:**
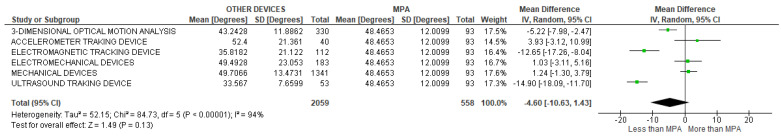
Mean, standard deviation, total of measures and 95% confidence interval comparing rRR of all other tool types with mobile phone applications.

**Figure 46 sensors-22-03042-f046:**
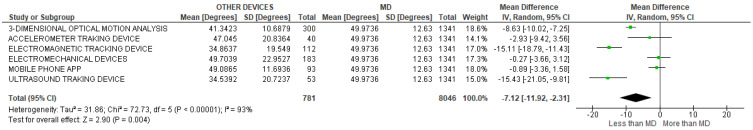
Mean, standard deviation, total of measures and 95% confidence interval comparing rLR of all other tool types with mechanical devices.

**Figure 47 sensors-22-03042-f047:**
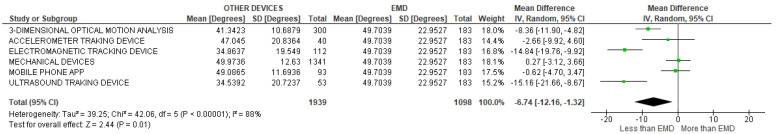
Mean, standard deviation, total of measures and 95% confidence interval comparing rLR of all other tool types with electromechanical devices.

**Figure 48 sensors-22-03042-f048:**
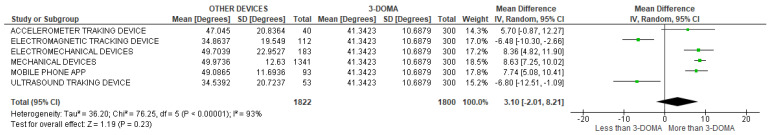
Mean, standard deviation, total of measures and 95% confidence interval comparing rLR of all other tool types with three-dimensional optical motion analysis.

**Figure 49 sensors-22-03042-f049:**
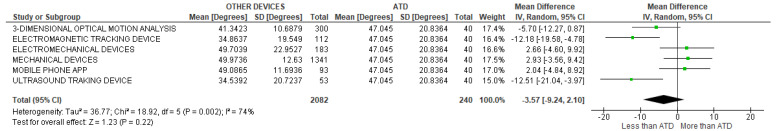
Mean, standard deviation, total of measures and 95% confidence interval comparing rLR of all other tool types with accelerometer tracking devices.

**Figure 50 sensors-22-03042-f050:**
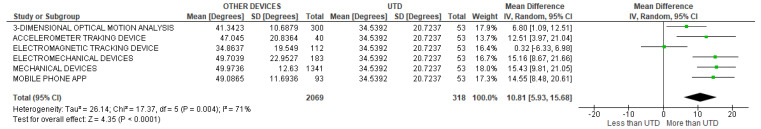
Mean, standard deviation, total of measures and 95% confidence interval comparing rLR of all other tool types with ultrasound tracking devices.

**Figure 51 sensors-22-03042-f051:**
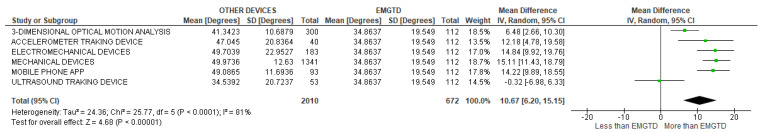
Mean, standard deviation, total of measures and 95% confidence interval comparing rLR of all other tool types with electromagnetic tracking devices.

**Figure 52 sensors-22-03042-f052:**
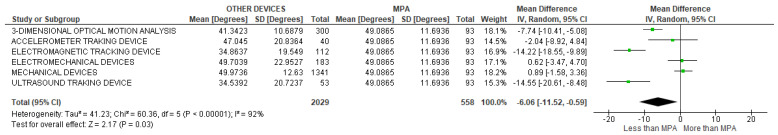
Mean, standard deviation, total of measures and 95% confidence interval comparing rLR of all other tool types with mobile phone applications.

**Figure 53 sensors-22-03042-f053:**
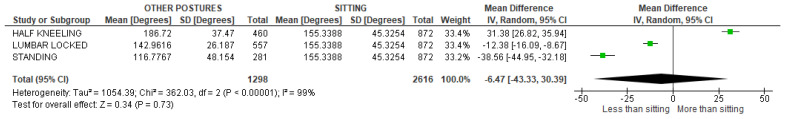
Mean, standard deviation, total of measures and 95% confidence interval comparing the measured rSSR of the other positions with the sitting position.

**Figure 54 sensors-22-03042-f054:**
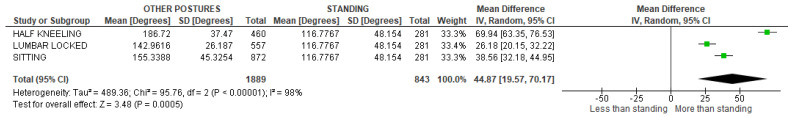
Mean, standard deviation, total of measures and 95% confidence interval comparing the measured rSSR of the other positions with the standing position.

**Figure 55 sensors-22-03042-f055:**
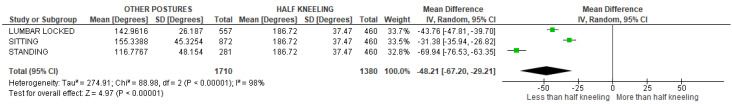
Mean, standard deviation, total of measures and 95% confidence interval comparing the measured rSSR of the other positions with the half kneeling position.

**Figure 56 sensors-22-03042-f056:**
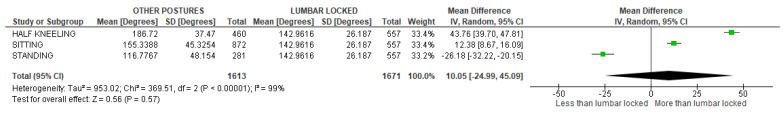
Mean, standard deviation, total of measures and 95% confidence interval comparing the measured rSSR of the other positions with the lumbar locked position.

**Figure 57 sensors-22-03042-f057:**
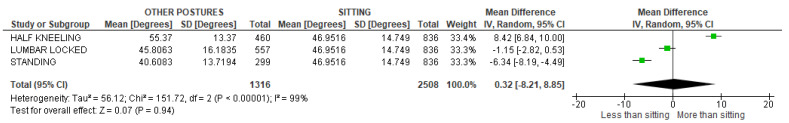
Mean, standard deviation, total of measures and 95% confidence interval comparing the measured rRR of the other positions with the sitting position.

**Figure 58 sensors-22-03042-f058:**
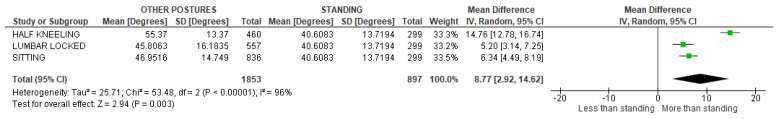
Mean, standard deviation, total of measures and 95% confidence interval comparing the measured rRR of the other positions with the standing position.

**Figure 59 sensors-22-03042-f059:**
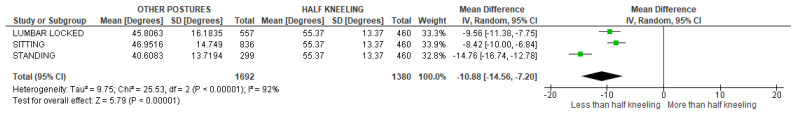
Mean, standard deviation, total of measures and 95% confidence interval comparing the measured rRR of the other positions with the half kneeling position.

**Figure 60 sensors-22-03042-f060:**
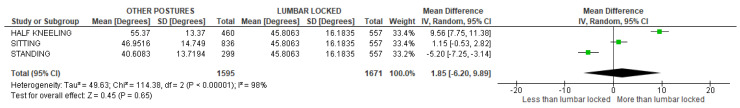
Mean, standard deviation, total of measures and 95% confidence interval comparing the measured rRR of the other positions with the lumbar locked position.

**Figure 61 sensors-22-03042-f061:**
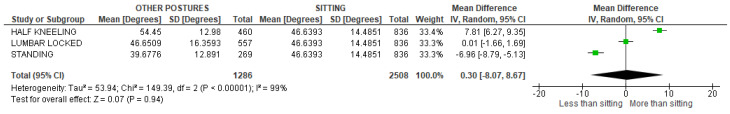
Mean, standard deviation, total of measures and 95% confidence interval comparing the measured rLR of the other positions with the sitting position.

**Figure 62 sensors-22-03042-f062:**
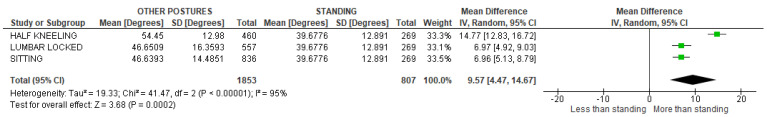
Mean, standard deviation, total of measures and 95% confidence interval comparing the measured rLR of the other positions with the standing position.

**Figure 63 sensors-22-03042-f063:**
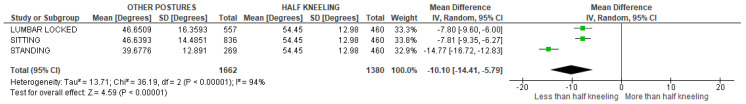
Mean, standard deviation, total of measures and 95% confidence interval comparing the measured rLR of the other positions with the half kneeling position.

**Figure 64 sensors-22-03042-f064:**
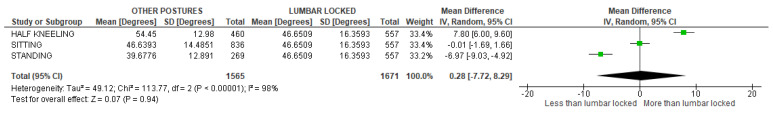
Mean, standard deviation, total of measures and 95% confidence interval comparing the measured rLR of the other positions with the lumbar locked position.

**Table 1 sensors-22-03042-t001:** Classification of the tools analyzed.

Type of Device	Device
Mechanical devices (MD)	INCLINOMETER, LIQUID GONIOMETER, GONIOMETER and BASELINE BUBBLE INCLINOMETER.
Electromechanical devices (EMD)	SPINAL MOUSE, VALEDOSHAPE, ACUMAR DI and ACCUMASTER.
Three-dimensional optical motion analysis (3-DOMA)	EXPERT VISION, 4-CAMERA AND SPHERICAL REFLECTIVE MARKERS, MET-SPOS, DIGITAL CAMERA AND SPHERICAL REFLECTIVE MARKERS, LATERAL DIGITAL PHOTOGRAPHS, OLYMPUS CAMERA AND PYRAMIDAL REFLECTIVE MARKERS, REFLECTIVE MARKERS AND CAMERA, VICON MX, POWERSHOT and OPTITRACK.
Accelerometer tracking device (ATD)	3A SENSOR STRING, X-SENS MTX and HALO.
Ultrasound tracking device (UTD)	CMS 20 ZEBRIS and POLHEMUS SYSTEM.
Electromagnetic tracking device (EMTD)	FASTRAK and FLOCK OF BIRDS.
Mobile phone app (MPA)	CLINOMETTER APP and COMPASS APP.

**Table 2 sensors-22-03042-t002:** Descriptive analysis of each study.

Author and Year	Objective	Methodology	Results and Conclusion
O’Gorman et al., 1987 [27]	Establish methods suitable for external measurement of thoracic spine measurement and to document normative values of thoracic mobility as well as sagittal plane posture in an aging population.	120 healthy adult female subjects were included in this study. Subjects were excluded from the study if they had thoracic spine pain, chronic neck pain, chronic low back pain, disease of the spine, or chest pathology.	This study has provided simple and repeatable methods of external measurement of thoracic kyphosis and movement suitable for a clinical setting. The changes of age were demonstrated.
Mellin G. et al., 1991 [28]	Compare reliability and range between spinal forward flexion in sitting and standing; extension in standing with and without the support and on an examination table; and lateral flexion in a free-standing position.	27 healthy subjects (10 male) chosen by chance among staff members of the Rehabilitation Foundation were included in this study. Subjects were excluded from the study if they had low back pain or if they were obese.	They improve the average range and repeatability but should not affect a comparison between positions because the testing sequence of each position was dispersed equally between the subjects.
Crawford H.J. et al., 1993 [29]	Investigate if the angle of “normal” thoracic kyphosis was related to the range of available arm elevation, to document the range of thoracic extension used in this action and its percentage of total available extension range.	60 healthy younger adult and healthy older female subjects were included in this study. Subjects were excluded from the study if they had thoracic spine pain, shoulder pain, scoliosis, chest conditions such as asthma, bronchitis, and emphysema, or conditions that may affect posture and movement.	Both normal young and older subjects use a high proportion of their thoracic extension range during bilateral arm elevation. Increased thoracic mobility in younger subjects is related to a large range of arm elevation, whereas an increased kyphosis in older adults is related to a reduced range of arm elevation.
Willems J.M. et al., 1996 [30]	Provide preliminary data on three-dimensional thoracic spine kinematics measured in vivo.	60 healthy young adult subjects (30 male) were included in this study. Subjects were excluded from the study if they had a history of thoracic spine pain or injury, a history of thoracic surgery, or a history of scoliosis.	This study has provided some preliminary data of three-dimensional thoracic kinematics in vivo. Axial rotation is the dominant movement of the thoracic region followed by a sagittal and coronal plane motion.
Gilleard W. et al., 2002 [25]	Investigate the effects of pregnancy on the kinematics of the trunk segments during seated and standing forward flexion, side to side flexion, and seated axial rotation and compare it with control subjects.	9 healthy maternal primiparous and multiparous subjects and 12 nulliparous subjects were included in this study.	The maternal subjects were similar to the control subjects in early pregnancy and at 8 weeks post-birth. In late pregnancy, the maternal subjects use strategies to minimize the effects of anatomical changes due to pregnancy.
Mannion A.F. et al., 2004 [31]	Assess the reliability of one of these types of devices, The Spinal Mouse.	20 healthy volunteers Subjects (9 male) were included in this study. Subjects were excluded from the study if they had any low back pain at the time of testing or had experienced so within the preceding 2 weeks.	For global regions of the spine, the Spinal Mouse delivered consistently reliable results for standing curvatures and range of motion both within and between days also between investigators.
Post R.B. et al., 2004 [32]	Test the spinal Mouse inter-rater reliability as well as judge the device on its merits in clinical practice.	111 subjects (75 male) were included in this study, 42 healthy subjects and 69 had sustained a spinal fracture. All spinal fracture subjects sustained their fracture at least 5 years previously and none of them had a neurological deficit.	The Spinal Mouse seems to be a good, reliable device for measuring sagittal spinal ROM, as tested inter-rater reliability. Measuring intersegmental RoM does not seem to be a reliable tool.
Holmström E. et al., 2005 [33]	Evaluate the effects on muscle stretchability, joint flexibility, muscle strength, and endurance in construction workers after a period of mourning warming-up exercise program of 3 months.	57 male construction workers healthy subjects were included in this study. Subjects were excluded from the study if they did not work at construction during the last 12 months or had diseases or symptoms in the examinations.	Thoracic and lower back flexion mobility increased after a period of the morning warming-up program and differed significantly from the controls. The endurance decrease in the control group and muscular strength was not affected.
Edmondston S.J. et al., 2007 [34]	Use an optical motion analysis system to examine ranges of axial rotation and coupled axial rotation of the mid thorax in asymptomatic subjects and to determine whether these patterns of coupled movement are influenced by the posture.	52 healthy subjects (25 male) were included in this study. Subjects were excluded from the study if they had conditions that may have affected the mobility of the thoracic spine such as trauma or surgery to the spine, spinal deformities, rheumatic disorders, or current thoracic pain.	In asymptomatic subjects, the rotational mobility of the thorax and the coupled lateral flexion are dependent on the posture from which the movement is initiated.
Tedereko P. et al., 2007 [26]	Present a prototypic station for active thoracic and lumbar ROM measurement with strict stabilization of the pelvis and lower limbs, analyze the repeatability, and analysis the neutral position reproducibility during the measurement.	12 healthy subjects (4 male) were included in this study. Subjects were excluded from the study if they had a history of musculoskeletal disorder, postural abnormalities, and no pain in the examination.	Validated spinal measurements of active range of motion are useful in the monitoring of patients with musculoskeletal disorders. Determination of reference values of normal thoracic and lumbar range of motion is problematic because of discrepancies between measurement protocols. Immobilization of pelvis and lower limbs improves the repeatability of assessment of the thoracic and lumbar range of active motion and the reproducibility of the neutral position.
Hsu C.J. et al., 2008 [35]	Evaluate the 3D movement patterns of the spine and measure the ROM in healthy adults using an electromagnetic tracking device, and to analyze the relative contribution of the thoracic spine, the lumbar spine, and the hip to trunk movements.	18 healthy male adult subjects were included in this study after signing informed consent.	With the electromagnetic tracking device, it is relatively simple and reliable to do a 3D dynamic measurement of the trunk movement objectively.
Mika A. et al., 2009 [36]	Determine whether the physical activity levels of postmenopausal women were associated with bone mineral density, the severity of thoracic kyphosis, and range of spinal motion.	189 healthy female subjects were included in this study. Subjects were excluded from the study if they had a chronic disease or other conditions which may influence muscle strength such as spondylarthrosis, rheumatoid arthritis, or acute back pain at the time of the evaluation.	Moderately active women had a better range of spinal motion than sedentary women, but they did not differ significantly in the severity of kyphosis and bone mineral density. This study supports the importance of physical activity in postmenopausal women with bone loss.
Kasukawa Y. et al., 2010 [37]	Evaluate differences in spinal kyphotic angle, spinal mobility, muscle power, and postural imbalance in elderly people with or without a history or fear of falls.	92 elderly subjects (23 male) who underwent a medical checkup for Musculoskeletal disorders from 2003 to 2007. Subjects were excluded from the study if they had neurological disorders.	The study reveals a relationship between spinal factors and falls.
Theisen C. et al., 2010 [38]	Compare the ROM of the thoracic spine in the sagittal plane in patients with outlet impingement syndrome and patients with no shoulder pathology.	78 adult subjects (46 male) were included in this study, 39 (23 male) with shoulder impingement, and 39 (23 male) healthy. Subjects in the impingement group were excluded from the study if they had concomitant pathologic conditions of the shoulder. The healthy subject group was excluded if they had any problem, pathology, or pain.	The use of ultrasound topometry shows altered sagittal mobility of the thoracic spine in patients with an outlet impingement syndrome of the shoulder compared with patients who had no shoulder pathology.
Heneghan N.R. et al., 2010 [39]	Describe soft-tissue artifact as a first attempt in quantifying this unknown source of measurement error during functional movements in the thoracic spine and to evaluate whether there was any association between the ranges of thoracic motion and the amount of skin displacement.	30 healthy subjects (14 male) were included in this study. Subjects were excluded from the study if they had previous neuromusculoskeletal spine conditions or who had scarring from abdominal surgeries.	This study describes soft-tissue artifacts during thoracic axial rotation and single-arm elevation using ultrasound imaging of bone and motion analysis to quantify the range of motion. The region of the greatest soft-tissue artifact was found in the mid-thoracic during axial rotation.
Imagama S. et al., 2011 [40]	Evaluate age-related changes in the lumbar spine, sagittal balance, spinal mobility, and back muscle strength in middle-aged and elderly males, and determine the relationship with quality of life	100 male healthy subjects were included in this study. Subjects were excluded from the study if they had a history of spinal surgery, history of spinal compression fracture or if they did not agree to the study	Quality of life of middle-aged and elderly males is related to factors such as sagittal balance, lumbar lordosis angle, spinal ROM, and back muscle strength.
Edmondston S.J. et al., 2011 [41]	Examine the global and regional extension mobility of the thoracic spine in young, asymptomatic adults, using the habitual standing thoracic kyphosis as a reference from which to define the ROM and to evaluate the influence of the thoracic kyphosis on the thoracic extension in standing.	40 healthy subjects (20 male) were included in this study. Subjects were excluded from the study if they had a history of spine pain in the previous 3 months, chronic respiratory disorders, and visually detected frontal plane deformities of the spine.	About the standing kyphosis, the sagittal mobility of the thoracic spine in young asymptomatic adults is relatively equal in flexion and extension. The magnitude of the thoracic kyphosis was associated with the end range extension position but not with the ROM toward an extension.
Edmondston S.J et al., 2012 [42]	Examine the range of thoracic spine extension motion in a group of young, asymptomatic subjects and compare the radiologically derived measurements with those obtained using photographic analysis and examine the relationship between the magnitude of neutral thoracic kyphosis and the range of thoracic spine extension motion.	14 healthy male subjects were included in this study. All subjects were university staff or students who volunteered to participate and were recruited for 2 months. Subjects were excluded from the study if they had thoracolumbar scoliosis, chronic respiratory disorders, or spinal pain requiring treatment in the previous 3 months.	The method has been used to demonstrate considerable variability in thoracic spine extension in asymptomatic spine extension. Radiographic measurements were moderately correlated with angular photographic measurements.
Fölsch C. et al., 2012 [43]	Investigate the test–retest reliability of the CMS 20 ultrasound analysis system in the measurement of kyphosis angle, end-range flexion, and end-range extension of the thoracic spine.	28 healthy subjects (14 male) were included in this study. Subjects were excluded from the study if they had pre-existing disease of the spine or pain within the previous year, pain during the examination, or failure to obey the instructions.	Ultrasound measurement analysis of static kyphosis angle of the thoracic spine in a sitting position provided good test–retest reliability. The ICC estimates were less for measurements of end-range flexion and even lower for the end-range extension. The high standard error of measurements and deviation differences seem to make this measurement unsuitable for motion analysis of the thoracic spine.
Johnson K.D. et al., 2012 [44]	Identify the most reliable techniques to measure thoracic spine rotation in healthy adults.	46 healthy subjects (15 male) were included in this study. Subjects were excluded from the study if they had any pathologic condition of the spine, rib, shoulder, hip, or knee within the past 6 months, a history of scoliosis, a rheumatologic or respiratory condition, or any chance or pregnancy.	Our results indicate that the 5 techniques can be measured reliably by the same clinician within a day and between days.
Edmondston S.J. et al., 2012 [45]	Measure thoracic spine extension motion during bilateral arm elevation in asymptomatic male subjects using functional radiographic analysis and validity of photographic measurements of thoracic extension motion through comparison with the radiological measurements.	21 healthy male subjects were included in this study. Subjects were excluded from the study if they had thoracolumbar scoliosis, Scheuermann disease, a history of spinal or shoulder pain in the previous 3 months, a body mass index greater than 25 kg/m^2^, radiographic or computerized tomography in the previous 12 months, or chronic respiratory disorders.	Functional radiographic analysis was used to measure the extension motion of the thoracic spine associated with bilateral arm elevation. When referenced to the thoracic kyphosis measured in the neutral standing position, the mean range of thoracic extension in the end range was 12.8 degrees with considerable variability among participants.
Wang H.J. Et al., 2012 [46]	Provide further evidence about the change of trunk mobility and the relationship between spinal curvature and balance and balance disorder, especially for the different types of global spine deformity in a Chinese population.	476 elderly women subjects with and without osteoporosis were included in this study. Subjects were excluded from the study if they had a neurologic or musculoskeletal disease.	The present study classified and compared the mobility and the curvature in a Chinese population based on the entire spinal alignment.
Benjamin-Hidalgo P.E. et al., 2012 [47]	Evaluate the intra-examiner reliability of active trunk motion measurements in healthy subjects and those which chronic low back pain, to study the responsiveness of the model and to determine the sensitivity and specificity of ROM and speed measurements during active trunk movement.	25 healthy subjects (10 male) and 25 subjects with low back pain (12 male) were included in this study. Healthy subjects were recruited voluntarily and had no incidence of low back pain in the 6 months before the experiment.	The quantitative analysis of kinematic motion patterns in subgroups of patients with chronic low back pain during trunk movements in different directions is of major importance because it can help clinicians to identify motion patterns that may contribute to chronic low back pain disorders and target interventions according to the quality of movement. The kinematic spine model and standardized protocol including 7 trunk motion tasks demonstrated good to excellent reliability.
Tsang S.M. et al., 2013 [48]	Examine the contribution and inter-regional coordination of the cervical and thoracic spine during active neck movements in a group of asymptomatic participants. This study will fill the knowledge gaps identified in this review, providing useful kinematic data of healthy participants which may help clinicians evaluate the neck mobility of patients with neck pain.	34 healthy subjects (10 male) were included in this study. Subjects were excluded from the study if they had any limitation in performing pain-free neck movements actively, or had any orthopedic, neurological, or vestibular conditions.	The present study showed that the upper thoracic spine contributes significantly to overall neck mobility, although the extent depends on the direction of neck movement. The inter-regional coordination between the cervical and thoracic spine during active neck movements was found to be high.
Battaglia G. et al., 2014 [49]	Investigate the changes in spinal ROM after an 8-week flexibility training program in elderly women, modulating the volume (sets and repetitions) of workload training.	37 healthy women subjects were included in this study. Subjects were included if they were over 60 years old, could provide informed consent, had a medical certificate attesting to their cognitive and physical suitability to participate in an experimental study, and were physically active.	In conclusion, our findings indicate that the flexibility training protocol performed for 8 weeks could improve spinal ROM in elderly women. These data might be suitable for increasing knowledge about the methodology of geriatric gymnastics. This study showed that a specific workload pattern (set, repetitions, type of exercise) could increase spinal ROM in elderly women.
Wirth B. et al., 2014 [50]	Investigate whether patients with chronic neck pain differ from healthy controls in terms of the thoracic spine and chest mobility and whether these parameters correlate positively with respiratory and neck function	19 healthy subjects (7 male) and 10 subjects with chronic neck pain (7 male) were included in this study. Subjects were excluded from the study if they had spinal fracture or surgery or neurological or inflammatory pathology.	Thoracic spine and chest mobility were related only to MVV and not to the maximal respiratory pressures, the finding of the relationship to all cervical motions is of clinical importance.
Çelenay ŞT. et al., 2015 [51]	Investigate the effects of postural education on posture and mobility, and assess and compare the effects of electrotherapy, exercise, biofeedback trainer in addition to postural education in university students.	96 healthy subjects (49 male) were included in this study. Subjects were excluded from the study if they had a systemic pathology including inflammatory disease; having a musculoskeletal injury, trauma, pathology, or structural deformity related to spine and extremities; or having active intervention including corticosteroid or any medication in the last 3 months.	Thoracic Spinal Stabilization Exercises were an effective and superior intervention on improving thoracic and lumbar spinal posture and mobility of university students. However, postural education itself was effective to change neither spinal posture nor mobility.
Talukdar K. et al., 2015 [52]	Investigate the role of upper-body rotational power and thoracic/hip mobility on cricket ball–throwing velocity.	11 male professional cricket players and 10 under-19 club-level cricketers were included in this study. Subjects were excluded from the study if they had assisted physiotherapists in the 2 months before or had any major musculoskeletal injury.	Significant differences were observed between fast and slow throwers regarding the chop (work and force) but not for the lift.
Alqhtani R.S. et al., 2015 [53]	Investigate the reliability of a novel motion analysis device for measuring the regional breakdown of spinal motion and describing the relative motion of different segments of the thoracolumbar spine.	18 healthy male subjects were included in this study. Subjects were excluded from the study if they had any spinal surgery, neurologic, or rheumatological disorders, or any disorder affecting the cervical, thoracic or lumbar region.	This multi-accelerometer system demonstrated excellent reliability and small errors to provide a viable and, largely practical, method of assessing multiregional clinical spinal motion.
Hajibozorgi M. et al., 2016 [15]	Measure total (T1–T12), lower (T5–T12) and upper (T1–T5) thoracic, lumbar (T12–S1), pelvis, and total trunk ROMs and their movement rhythms in the sagittal plane and	40 young healthy male student subjects with no history of back surgery or recent back, hip, or knee complications were included in this study.	The thoracic spine ROM during forwarding trunk flexion could have implications in patient discrimination and biomechanical models. Inertial tracking devices allow for straightforward measurement of spinal ROMs. Thoracic sagittal ROM, mostly provided by movements from the lower (T5–T12) motion segments, was significantly smaller than that of the lumbar.
Schinkel-Ivy A. et al., 2016 [54]	Provide a preliminary indication of the relationships between breast size and spine motion and muscle activation variables in a sample of healthy young females.	15 university-aged female subjects with all right dominant and without back pain were included in this study.	The results of the present study indicated that for a sample of young, healthy females across a range of breast sizes, increasing breast size was related to more extended Head and Trunk angles, as well as greater Thoracic flexion angles during flexion postures.
Furness J. et al., 2016 [55]	Develop a reliable method to quantify thoracic mobility in the sagittal plane; assess the reliability of an existing thoracic rotation method and quantify thoracic mobility in an elite male surfing population.	57 healthy subjects (26 male) and 15 elite male surfers were included in this study. Subjects were excluded from the study if they had any cute or chronic spinal pathology in the past 3 months.	This study has illustrated reliable methods to assess the thoracic spine in the sagittal and horizontal planes. It has also quantified ROM in a surfing cohort; identifying thoracic rotation as a key movement.
Mazzone B. et al., 2016 [56]	Compare spine kinematics during prone extension in subjects with and without low back pain. Exploratory analyses were conducted to investigate differences among low back pain subgroups.	17 healthy subjects (7 male) and 18 with low back pain (7 male) were included in this study. Subjects were excluded from the study if they were pregnant or had a history of serious spinal or other medical conditions except for low back pain for the study group.	There were no differences in overall trunk extension kinematics between subjects with and without low back pain. However, the distribution of movement differed between groups. Subjects with LBP displayed less low lumbar spine extension than subjects without low back pain.
Zafereo J. et al., 2016 [57]	Determine the reliability of using a skin-surface device to measure global and segmental thoracic and lumbar spine motion in participants with and without low back pain (LBP) and to compare global thoracic and lumbar motion between the 2 groups.	20 healthy subjects (5 male) and 20 subjects with low back pain (5 male) were included in this study. Subjects were excluded from the study if they had the presence of red flag signs or symptoms such as tumor, infection or cauda equina syndrome, previous spinal surgery, presence of spinal fracture, pregnancy, unable to complete segmental mobility, or if they were older than 75 years old.	Global thoracic and lumbar end-range motion measurement using a skin-surface device has acceptable reliability for participants with LBP. Reliability for segmental end-range motion measurement was only acceptable for lumbar flexion in participants with LBP.
Morais N. et al., 2016 [58]	Explore whether postural alignment and mobility variables of the upper quadrant are associated with changes in pulmonary function and compare such variables between patients with chronic obstructive pulmonary disease (COPD) and healthy individuals.	15 healthy subjects (7 male) and 15 subjects with COPD (7 male) were included in this study. COPD subjects were included if they were ≥18 years old, clinically stable over the past month, living in the community, able to walk, and able to follow instructions and were excluded if they had thoracic or abdominal surgery, recent musculoskeletal injury, or cardiovascular disorders	Patients with COPD presented impaired pulmonary function associated with pectoralis minor muscle length and mobility of the upper quadrant possibly as musculoskeletal adaptations to the chronic respiratory condition.
Rast F. et al., 2016 [59]	Quantify and compare the between-day reliability of trunk kinematics, when using an optoelectronic system and skin markers.	20 healthy subjects (10 male) were included in this study. Subjects were excluded from the study if they were overweight (body mass index ≥25 kg/m^2^)	The additional markers and the point cloud algorithm used in this study did not improve the between-day reliability of trunk kinematics but resulted in different magnitudes of axial rotation angles. Furthermore, using a reference trial to define neutral position was found to be more reliable for analysis of frontal and transverse plane movements, whereas the definition by anatomical landmarks was more reliable for sagittal plane movements.
Ishikawa Y. et al., 2017 [60]	Investigate the relationships of total-body inclination, including the cranium, and sagittal alignment and mobility of the spine and lower extremities to quality of life (QOL) and falls, and to clarify which types of alignment and mobility of the spine and lower extremities correlate with QOL and falls among community-dwelling individuals.	110 healthy subjects (41 male) were included in this study. All participants were able to walk and displayed no neurologic, scoliotic, or metabolic disorders related to spinal alignment.	Our results suggest that decreased extension range of motion of the lumbar spine is one of the most significant factors for falling. Screening those patients who demonstrate less ability to extend the spine and prescribing exercise therapy to regain extension mobility may reduce the incidence of falls. Forward-stooped posture and knee flexion deformities are associated with reduced QOL.
Bucke J. et al., 2017 [11]	Explore the criterion and concurrent validity of a digital inclinometer (DI) and iPhone Clinometer app for measuring thoracic spine rotation using the heel-sit position.	23 healthy subjects (14 male) were included in this study. Subjects were excluded from the study if they had experienced a neuromusculoskeletal spine problem within the 12 months before the study, rheumatologic condition, current or chronic respiratory condition, were pregnant or were unable to adopt the heel-sit position.	The DI and iPhone provided valid measures of thoracic spine rotation in the heel-sit position. Both can be used in clinical practice to assess thoracic spine rotation, which may be valuable when evaluating thoracic dysfunction.
Roghani T. et al., 2017 [61]	The purpose of this study was to investigate the interrater reliability of a skin-surface instrument (Spinal Mouse, Idiag, Voletswil, Switzerland) in measuring standing sagittal curvature and global mobility of the spine in older women with and without hyperkyphosis.	18 healthy women subjects and 20 women subjects with hyperkyphosis were included in this study.	Our study reports very high interrater reliability of the Spinal Mouse for the measurement of spinal curvature and mobility in older women with and without hyperkyphosis. Although the Spinal Mouse cannot replace the gold standard evaluation of spinal curvature with lateral spinal radiographs, our study suggests that this device can be used to reliably assess spinal curvature and mobility in older women with and without spinal deformities.
Hwang D. et al., 2017 [8]	Measure the accurate angle of thoracic rotation and determine which measurement device is the most reliable among the four commonly used by the therapist and suggest the most reliable and convenient way to measure thoracic ROM.	40 healthy subjects (20 male) were included in this study. Subjects were excluded from the study if they had rheumatic disease, pain, congenital or acquired disease around the thoracic spine or if they were pregnant.	The use of the goniometer, bubble inclinometer, dual inclinometer, and smartphone clinometer for measurements in the lumbar locked posture are reliable and valid non-radiologic measures of thoracic rotational ROM in healthy adults.
Narimani M. et al., 2018 [62]	Measure T1, T5, T12, total (T1–T12) thoracic, lower (T5–T12) and upper (T1–T5) thoracic, lumbar (T12–S1), and pelvis primary and coupled ROM in all anatomical planes and directions (flexion, extension, left/right lateral bending, and left/right axial rotation) during unconstrained standing posture in healthy individuals.	22 young healthy male subjects were included in this study. Subjects were excluded from the study if they had recent back, hip, or knee complications.	Pelvis, the lumbar, and thoracic spine had different/varying contributions/rhythms to generate total trunk (T1) movement, both within and between planes. The pattern of the coupled motions was inconsistent between subjects, but side bending was generally associated with twisting to the same side at the thoracic spine and the opposite side at the lumbar spine.
Heneghan N.R. et al., 2018 [63]	Investigate the influence of sedentary behavior on thoracic spine mobility. Investigate the influence of physical activity on thoracic spine mobility. To evaluate whether a relationship exists between duration of sitting and physical activity and thoracic mobility.	96 healthy asymptomatic subjects (35 male) were included in this study. Subjects were included in the study if they were 18–30 years. They were excluded if they had current or previous neuromusculoskeletal spine condition, rheumatoid arthritis, current or chronic respiratory conditions, pregnancy, current or knee pathology, or were unable to adopt the heel-sit position.	This study provides evidence of reduced thoracic mobility in individuals who spend >7 h a day sitting and <150 min of physical activity a week. With observed associations between thoracic mobility and exercise and sitting duration, further research is now required to explore the possible causal relationship between physical activity behaviors on spinal musculoskeletal health and subsequently their relationship to spinal complaints.
Mousavi S.J. et al., 2018 [64]	Measure thoracic kyphosis (TK), lumbar lordosis (LL), and pelvic tilt (PT), as well as three-dimensional spine flexion, extension, lateral bending, and axial rotation ROMs, with three-dimensional marker clusters on the spine, and to determine the between-session reliability of these measurements.	19 healthy subjects (11 male) were included in this study. Subjects were excluded from the study if they had recent back pain, history of spinal surgery, traumatic fracture, thoracic deformity, or conditions that affect balance, movement, or ability to stand.	this study demonstrates that optoelectronic motion capture measurements afford objective, quantitative and reliable data on a patient’s posture and kinematics. Importantly, we demonstrate that reliable data can be obtained with a reasonable number of trial repetitions for most outcomes. In addition, motion capture allows for three-dimensional and dynamic outcomes to be assessed, which would not be possible with standard diagnostic approaches such as radiographic studies.
Furness J. et al., 2018 [65]	Determine the reliability (intra-rater and inter-rater) and validity of the Compass app when assessing thoracic spine rotation ROM in healthy individuals.	30 healthy subjects (10 male) were included in this study. Subjects were excluded from the study if they were currently experiencing back or trunk pain, had any back injury within 6 weeks before testing, had a history of spinal surgery, were younger than 18 years of age, or refuse to give informed consent.	This study reveals that a compass app is a reliable tool for measuring thoracic spine rotation which produces greater reproducibility of measurements both within and between raters than a universal goniometer (UG). As a significant positive correlation exists between the Compass app and UG, this supports the use of either device in clinical practice as a reliable and valid tool to measure thoracic rotation.
Beaudette S.M. et al., 2019 [66]	Identify if distinct spine spatiotemporal movement strategies are utilized in a homogenous sample of young healthy participants.	51 healthy male subjects were included in this study. Subjects were excluded from the study if they had trunk or pelvic pain or any diagnosed allergies to adhesives.	Spatiotemporal spine flexion-extension patterns are not uniform across a population of young healthy individuals.
Schinkel-Ivy A. et al., 2019 [67]	Investigate the interaction between thoracic movement and lumbar muscle co-contraction when the lumbar spine was held in a relatively neutral posture.	30 healthy subjects (15 male) were included in this study. All participants were right-hand dominant and were asymptomatic.	Tasks with thoracic movement and a neutral lumbar spine posture resulted in increases in co-contraction within the lumbar musculature compared with quiet standing and maximum trunk range-of-motion tasks. Findings indicated an interaction between the 2 spine regions, suggesting that thoracic posture should be accounted for during the investigation of lumbar spine mechanics.
Welbeck A.N. Et al., 2019 [68]	Examine the differences in thoracic spine rotation in swimmers with and without scapular dyskinesis and the relationship between thoracic spine rotation and shoulder pain/dysfunction according to the Kerlan-Jobe Orthopedic Clinic (KJOC) score.	34 NCAA division 1 swimmer subjects (13 males) were included in this study. Subjects were included in the study if they were swimmers ranging in ages from 18 to 26 years old, currently on the roster of a varsity level college swimming team and cleared by medical personnel for full participation in training and competition.	In our cohort of NCAA Division 1 swimmers, no differences were found between swimmers with or without scapular dyskinesis and the extent of thoracic rotation. We found no correlation between thoracic rotation and the amount of self-reported pain and dysfunction experienced in the upper extremity.
Hunter D.J. et al., 2020 [69]	Investigate whether there is a relationship between Shoulder impingement syndrome and thoracic posture.	39 healthy subjects (19 male) and 39 subjects with shoulder pain (20 male) were included in this study. Subjects were excluded from the study if they had any back injury within 6 weeks before testing, had a history of spinal surgery, were younger than 18 years of age, or refuse to give informed consent.	Individuals with SIS had a greater thoracic kyphosis and less extension ROM than age and gender-matched healthy controls. These results suggest that clinicians could consider addressing the thoracic spine in patients with SIS.

RoM: Range of Movement.

**Table 3 sensors-22-03042-t003:** Characteristics of the included studies.

Authors and Year	Age Mean (SD)	Device	Type of Device	Posture	Gender Separated	RoM Plane
O’Gorman et al., 1987 [27]	48.83 (17.02)	INCLINOMETER	Mechanical device	Sitting	Yes	Sagittal Coronal
Mellin G. et al., 1991 [28]	30.6 (8.9)	INCLINOMETER	Mechanical device	Sitting and standing	No	Sagittal Coronal
Crawford H.J. et al., 1993 [29]	43.25 (21.71)	INCLINOMETER	Mechanical device	Sitting	Yes	Sagittal
Willems J.M. et al., 1996 [30]	21 (3)	FASTRAK	Electromagnetic tracking device	Sitting	No	Sagittal Coronal Transversal
Gilleard W. et al., 2002 [25]	28 (7)	EXPERT VISION	Three-dimensional optical motion analysis	Sitting and standing	Yes	Sagittal Coronal Transversal
Mannion A.F. et al., 2004 [31]	41.8 (7.75)	SPINAL MOUSE	Electromechanical device	Standing	No	Sagittal
Post R.B. et al., 2004 [32]	39.2 (18)	SPINAL MOUSE	Electromechanical device	Standing	No	Sagittal
Holmström E. et al., 2005 [33]	39.7 (13.6)	LIQUID GONIOMETER	Mechanical device	Standing	Yes	Sagittal
Edmondston S.J. et al., 2007 [34]	23.2 (5.2)	4-CAMERA AND SPHERICAL REFLECTIVE MARKERS	Three-dimensional optical motion analysis	Sitting	No	Transversal
Tedereko P. et al., 2007 [26]	31.6 (13.6)	MET-SPOS	Three-dimensional optical motion analysis	Standing	No	Sagittal Coronal Transversal
Hsu C.J. et al., 2008 [35]	31 (13)	FLOCK OF BIRDS ELECTROMAGNETIC TRACKING DEVICE	Electromagnetic tracking device	Standing	Yes	Sagittal Coronal Transversal
Mika A. et al., 2009 [36]	64.7 (9)	GONIOMETER	Mechanical device	Standing	Yes	Sagittal Coronal
Kasukawa Y. et al., 2010 [37]	72.9 (8.1)	SPINAL MOUSE	Electromechanical device	Standing	No	Sagittal
Theisen C. et al., 2010 [38]	56.1 (19.5)	CMS 20 ZEBRIS	Ultrasound tracking device	Sitting	No	Sagittal
Heneghan N.R. et al., 2010 [39]	23.83 (3.1)	POLHEMUS SYSTEM	Ultrasound tracking device	Sitting	No	Transversal
Imagama S. et al., 2011 [40]	70.2 (7.1)	SPINAL MOUSE	Electromechanical device	Standing	Yes	Sagittal
Edmondston S.J. et al., 2011 [41]	22.8 (3.2)	DIGITAL CAMERA AND SPHERICAL REFLECTIVE MARKERS	Three-dimensional optical motion analysis	Sitting and standing	Yes	Sagittal
Edmondston S.J et al., 2012 [42]	30.2 (7)	LATERAL DIGITAL PHOTOGRAPHS	Three-dimensional optical motion analysis	Standing	Yes	Sagittal
Fölsch C. et al., 2012 [43]	33 (14.8)	CMS 20 ZEBRIS	Ultrasound tracking device	Sitting	No	Sagittal
Johnson K.D. et al., 2012 [44]	23.6 (4.3)	GONIOMETER AND INCLINOMETER	Mechanical device	Sitting, half kneeling, and lumbar locked rotation test	No	Transversal
Edmondston S.J. et al., 2012 [45]	22.6 (3.2)	OLYMPUS CAMERA AND PYRAMIDAL REFLECTIVE MARKERS	Three-dimensional optical motion analysis	Standing	Yes	Sagittal
Wang H.J. et al., 2012 [46]	73.34 (6.98)	SPINAL MOUSE	Electromechanical device	Standing	Yes	Sagittal
Benjamin-Hidalgo P.E. et al., 2012 [47]	40 (11)	REFLECTIVE MARKERS AND CAMERA	Three-dimensional optical motion analysis	Sitting	No	Sagittal Transversal
Tsang S.M. et al., 2013 [48]	34.5 (9.08)	FASTRAK	Electromagnetic tracking device	Sitting	No	Sagittal Coronal Transversal
Battaglia G. et al., 2014 [49]	69.1 (7.14)	SPINAL MOUSE	Electromechanical device	Standing	Yes	Sagittal
Wirth B. et al., 2014 [50]	56.5 (9.9)	SPINAL MOUSE	Electromechanical device	Standing	No	Sagittal
Çelenay ŞT. et al., 2015 [51]	20.1 (1.1)	SPINAL MOUSE	Electromechanical device	Sitting and standing	No	Sagittal
Talukdar K. et al., 2015 [52]	23.8 (2.27)	GONIOMETER	Mechanical device	Sitting	Yes	Transversal
Alqhtani R.S. et al., 2015 [53]	30.6 (7.4)	3A SENSOR STRING	Accelerometer tracking device	Standing	Yes	Sagittal Coronal Transversal
Hajibozorgi M. et al., 2016 [15]	22.5 (1.8)	X-SENS MTX	Accelerometer tracking device	Standing	Yes	Sagittal
Schinkel-Ivy A. et al., 2016 [54]	22.8 (2.7)	VICON MX	Three-dimensional optical motion analysis	Standing	Yes	Sagittal
Furness J. et al., 2016 [55]	31.29 (11.2)	HALO AND INCLINOMETER	ACCELEROMETER TRACKING DEVICE and MECHANICAL DEVICE	Sitting and lumbar locked rotation test	No	Sagittal Transversal
Mazzone B. et al., 2016 [56]	25.6 (8.7)	VICON MX	Three-dimensional optical motion analysis	Standing	No	Sagittal
Zafereo J. et al., 2016 [57]	29.9 (10.18)	VALEDOSHAPE	Electromechanical device	Standing	No	Sagittal
Morais N. et al., 2016 [58]	66.8 (7.47)	POWERSHOT	Three-dimensional optical motion analysis	Standing	No	Sagittal
Rast F. et al., 2016 [59]	29.95 (8.5)	VICON MX	Three-dimensional optical motion analysis	Standing	No	Coronal Transversal
Ishikawa Y. et al., 2017 [60]	72.9 (7.72)	SPINAL MOUSE	Electromechanical device	Standing	No	Sagittal
Bucke J. et al., 2017 [11]	25.82 (4.28)	POLHEMUS SYSTEM, ACUMAR DI AND CLINOMETTER APP	Ultrasound tracking device, electromechanical device, and mobile phone app	Lumbar locked rotation test	No	Transversal
Roghani T. et al., 2017 [61]	63 (6)	SPINAL MOUSE	Electromechanical device	Standing	Yes	Sagittal
Hwang D. et al., 2017 [8]	22.5 (3.5)	GONIOMETER, BASELINE BUBBLE INCLINOMETER, INCLINOMETER, AND CLINOMETER APP	Mechanical devices and mobile phone app	Lumbar locked rotation test	No	Transversal
Narimani M. et al., 2018 [62]	24.8 (1)	X-SENS MTX	Accelerometer tracking device	Standing	Yes	Sagittal Coronal Transversal
Heneghan N.R. et al., 2018 [63]	21.2 (2.6)	ACUMAR DI (DIGITAL INCLINOMETER)	Electromechanical device	Standing	No	Transversal
Mousavi S.J. et al., 2018 [64]	47 (17)	VICON MX	Three-dimensional optical motion analysis	Standing	No	Sagittal Coronal Transversal
Furness J. et al., 2018 [65]	29.8 (8.9)	COMPASS APP AND GONIOMETER	Mobile phone app and mechanical device	Sitting	No	Transversal
Beaudette S.M. et al., 2019 [66]	24 (3.3)	OPTITRACK	Three-dimensional optical motion analysis	Standing	Yes	Sagittal
Schinkel-Ivy A. et al., 2019 [67]	23.9 (3.25)	VICON MX	Three-dimensional optical motion analysis	Standing	Yes	Sagittal Coronal Transversal
Welbeck A.N. et al., 2019 [70]	19.6 (1.2)	ACCUMASTER (DIGITAL INCLINOMETER)	Electromechanical device	Lumbar locked rotation test	Yes	Transversal
Hunter D.J. et al., 2020 [68]	55.7 (10.6)	INCLINOMETER	Mechanical device	Sitting	No	Sagittal

N: number of subjects, ROM: Range of motion, and H.S.: Healthy subjects.

**Table 4 sensors-22-03042-t004:** Number of healthy subjects in function of measurement realized.

Measurements	n	Male	Female
rFE	1092	528	564
rF	1292	464	828
rE	951	410	541
rSSLF	561	142	419
rRLF	569	152	417
rLLF	539	137	402
rSSR	858	419	439
rRR	876	434	442
rLR	846	419	427
Total	2365	1053	1312

n: number of subjects, rFE: range of Flexoextension, rF: range of Flexion, rE: range of Extension, rSSLF: range of Side-to-Side Lateral Flexion, rRLF: range of Right Lateral Flexion, rLLF: range of Left Lateral Flexion, rSSR: range of Side-to-Side Rotation, rRR: range of Right Rotation and rLR: range of Left Rotation.

**Table 5 sensors-22-03042-t005:** The number of studies according to the different ranges of movement and device used.

Plane Measurement	MD	EMD	3-DOMA	ATD	UTD	EMGTD	MPA	Total Measures
rFE	3	10	5	2	2	3	0	25
rF	6	8	6	4	2	3	0	29
rE	3	7	7	2	2	3	0	24
rSSLF	3	0	4	2	0	3	0	12
rRLF	3	0	4	2	0	3	0	12
rLLF	3	0	3	2	0	3	0	11
rSSR	7	3	6	2	2	3	3	26
rRR	7	3	6	2	2	3	3	26
rLR	7	3	5	2	2	3	3	25
Total	42	34	46	20	12	27	9	190

MD: mechanical device, EMD: electromechanical device, 3-DOMA: three-dimensional optical motion analysis, ATD: accelerometer tracking device, UTD: ultrasound tracking device, MPA: mobile phone application, rFE: range of Flexoextension, rF: range of Flexion, rE: range of Extension, rSSLF: range of Side to Side Lateral Flexion, rRLF: range of Right Lateral Flexion, rLLF: range of Left Lateral Flexion, rSSR: range of Side to Side Rotation, rRR: range of Right Rotation, and rLR: range of Left Rotation.

**Table 6 sensors-22-03042-t006:** Number of studies according to the different measurements postures and devices used.

Device	Sitting	Standing	Half Kneeling	Lumbar Locked Rotation Test
MD	7	3	3	5
EMD	1	12	0	2
3-DOMA	4	12	0	0
ATD	1	3	0	0
UTD	3	0	0	1
EMGTD	1	2	0	0
MPA	1	0	0	2
Total	18	32	3	10

MD: mechanical device, EMD: electromechanical device, 3-DOMA: three-dimensional optical motion analysis, ATD: accelerometer tracking device, UTD: ultrasound tracking device, EMGTD: electromagnetic tracking device and MPA: mobile phone application.

**Table 7 sensors-22-03042-t007:** Demographic data of the selected subjects depending on the type of device used.

Posture Measure	n	Years (SD)	BMI (SD)
MD	777	44.18 (19.58)	26.27 (4.26)
EMD	749	44.26 (23.87)	23.78 (4.10)
3-DOMA	455	27.57 (11.26)	23.80 (3.19)
ATD	89	26.65 (7.69)	24.03 (3.32)
UTD	124	36.84 (18.74)	22.87 (3.48)
EMGTD	94	28.07 (7.47)	23.79 (3.95)
MPA	78	28.07 (7.47)	23.75 (3.15)
Total	2365	39.24 (20.64)	24.44 (3.81)

n: number of subjects, SD: standard deviation, MD: mechanical device, EMD: electromechanical device, 3-DOMA: three-dimensional optical motion analysis, ATD: accelerometer tracking device, UTD: ultrasound tracking device, EMGTD: electromagnetic tracking device and MPA: mobile phone application.

**Table 8 sensors-22-03042-t008:** Demographic data of the selected subjects depending on the type of position measure.

Posture Measure	n	Years (SD)	BMI (SD)
SIT	735	33.57 (17.03)	23.46 (3.27)
ST	1242	44.59 (22.52)	24.60 (3.76)
HN	77	23.5 (4.35)	24.40 (3.30)
LL	311	24.06 (6.36)	24.04 (2.98)
Total	2365	39.24 (20.64)	24.44 (3.81)

n: number of subjects, SD: standard deviation, SIT: sitting, ST: standing, HN: half-kneeling, and LL: lumbar locked.

**Table 9 sensors-22-03042-t009:** Mean, standard deviation, total of measures and 95% confidence interval comparing standing position with sitting position in rFE, rF, rE, rssLF, rRLF and rLLF.

Posture Measure	Standing			Sitting			Mean Difference
	Mean [Degrees]	SD [Degrees]	Total	Mean [Degrees]	SD [Degrees]	Total	IV, Fixed, 95% CI
rFE	62.8485	52.0603	818	71.0636	34.3889	480	−8.22 [−12.93, −3.50]
rF	35.3205	25.6757	906	36.2647	27.815	488	−0.94 [−3.93, 2.04]
rE	13.8573	16.1804	633	20.0032	12.4703	388	−6.15 [−7.91, 4.38]
rssLF	104.9546	36.1702	441	78.2215	27.7242	250	26.73 [21.92, 31.55]
rRLF	33.2965	12.4733	423	23.88	10.2095	214	9.42 [7.69, 11.23]
rLLF	30.4491	11.1596	393	23.5919	9.7917	214	6.86 [5.14, 8.57]

rFE: range of flexoextension, rF: range of flexion, rE: range of extension, rssLF: range of side-to-side lateral flexion, rRLF: range of right lateral flexion and rLLF: range of left lateral flexion.

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
