# Peer review of "Analysis of the Active Measurement Systems of the Thoracic Range of Movements of the Spine: A Systematic Review and a Meta-Analysis"

_sensors, 2022, doi:10.3390/s22083042_

Round 1

Reviewer 1 Report

Title: Please address ‘a network meta-analysis’ in the title.

Abstract: The narrative is too lengthy and it is suggested to reduce the number of words.

  1. PEDro scale is designed to evaluate the randomized clinical trials studies, in this study the authors want to compare the different tools measuring the thoracic range of movements of the spine for healthy subjects, it’s seemed not to be suitable.
  2. In Page4, lines 30-31: For all abbreviations that appear for the first time, please write the full text for easy reading and understanding, such as rFE, rF, rE, rSSLF, rRLF, rLLF, rSSR, rRR, rLR, MD, EMD, 3-DOMA, ATD, UTD, EMGTD and MPA.
  3. The MINORs methodological rating scales presented in Appendix B, accord the evaluated score of items, how to get the score 7.50 [24], 8.75 [25], 9.38 [27], …in the Appendix?
  4. In Page 4, line37-38, The scale of heterogeneity was considered, whereby <25% indicates low, 25-75% medium, and >75% high heterogeneity. In the forest plots, all of them the I2 > 25%, so might be select the random effect model was better than fixed effect model.
  5. Figure 2-10 showed many types of devices, please provide the complete name of the device, and had 2 ‘Goniometer’ in Fig. 8-10, need to clarified.
  6. The authors lack to explain which devices were count to ‘mechanical device’, ‘electromechanical device’, ‘3-dimensional optical motion analysis’, ‘accelerometer tracking device’, ‘ultrasound tracking device’, ‘mobile phone application’.
  7. There are different types of measurement tools, and their rationales used in the measurements are different, how to establish a standardized performance protocol?

Reviewer 2 Report

This paper systematically summarizes and analyzes the current active measurement systems of the thoracic range of movements of the spine, and finds that different measurement protocols will lead to different results. It is a meaningful work. However, to enhance its readability and usefulness, the following revisions can be considered.

  1. P1 Line 17-45: The Objective, Method, Result and Conclusion in the abstract can be divide into separate paragraphs and annotate them with ordinal numbers, such as (1) Objective, which will enable readers to read more effectively.
  2. P1 Line 46-47: There are too many keywords, maybe you can simplify them, which will make the article position more accurately.
  3. P3 Line 7 and 16: “The following inclusion criteria were used:”, “The studies with the following standards were excluded:”

I think it would be better to explain the specific references of such standards.

  1. P3 Line 48: “…AND (thoracic)) AND (measure)))”

I think this paragraph is somewhat redundant and can be put in the appendix. In addition, it would be better to add a full stop at the end of the paragraph.

  1. P4 Line16 and 22: Indenting the first line of these two paragraphs will be more beneficial to readers.
  2. P4 Line 30: “The mean values and 95% confidence interval (CI) of rFE, rF, rE, rSSLF, rRLF, rLLF, rSSR, rRR and rLR 30were calculated for different type of device groups (MD, EMD, 3-DOMA, ATD, UTD, EMGTD and MPA)”

I think it is more convenient for readers to explain the meaning of abbreviations here.

  1. P5 Line 13:There is a problem with the layout here, as shown in the figure below. The picture name should be centered and the subtitle should be “3.2. Study characteristics” left aligned.
  2. P23 Table5 and P24 Table6: It is better to add the meaning of EMGTD in the notes.
  3. P25 Fig2: I think it's best to enlarge its size to align with Fig3 and Fig4 in proportion.
  4. P25 Fig3 and P27 Fig9: The picture ordinate has no dimension unit.
  5. P28 Table8-11: Some table names coincide with the pictures.

I think it's best to appropriately increase the up and down distance between them, as shown in the figure below.In addition, the abbreviated full name of the range of motion can be added to each table for the convenience of readers. For example: rF-E(range of Flexoextension).

  1. P44 Line 237: “The UTD and EMGTD measurements only…”

The sentence seems to be preceded by a space character.

  1. P45 Line 269: “If this is achieved, it could even be decisive in the joint assessment of the movement of the 269thoracic spine with the presence of pain…”

I think it is best to cite relevant literature to prove the reliability of this conclusion.

  1. This paper systematically summarizes and analyzes the current active measurement systems of the thoracic range of movements of the spine, and finds that different measurement protocols will lead to different results. Although the ideas and general experimental methods to solve this problem are put forward in the discussion part, there is no specific and feasible experimental design scheme. For example, can the author design a set of practical experimental scheme to prove this conclusion? Or will it be in the future study?

Round 2

Reviewer 1 Report

1. Because the content is too lengthy, it is recommended to delete Figures 2-10, after all, the relevant information is integrated and presented in the following forest plots.

2. Since the heterogeneity of the studies in Table 9-17 and 24-26 were too high, and it is necessary to use the random effect model to avoid overestimating the overall effect of test, not select the fixed effect model.

3. Tables 18-23 each have only one study and are not suitable for forest plots to present the results of a meta-analysis.

4. The reference numbers indicated in Tables 3 and Appendices B and C are inconsistent and need to be corrected.

Round 3

Reviewer 1 Report

All have been revised and there are no other comments.